# Nontargeted Metabolomics by High-Resolution Mass Spectrometry to Study the In Vitro Metabolism of a Dual Inverse Agonist of Estrogen-Related Receptors β and γ, DN203368

**DOI:** 10.3390/pharmaceutics13060776

**Published:** 2021-05-31

**Authors:** Sin-Eun Kim, Seung-Bae Ji, Euihyeon Kim, Minseon Jeong, Jina Kim, Gyung-Min Lee, Hyung-Ju Seo, Subin Bae, Yeojin Jeong, Sangkyu Lee, Sunghwan Kim, Taeho Lee, Sung Jin Cho, Kwang-Hyeon Liu

**Affiliations:** 1BK21 FOUR KNU Community-Based Intelligent Novel Drug Discovery Education Unit, College of Pharmacy and Research Institute of Pharmaceutical Sciences, Kyungpook National University, Daegu 41566, Korea; hjkopsty@gmail.com (S.-E.K.); wltmdqo2377@naver.com (S.-B.J.); uihyeon1112@naver.com (E.K.); lgm00179@naver.com (G.-M.L.); hlhl103@naver.com (H.-J.S.); bsb960908@naver.com (S.B.); duwls9902@naver.com (Y.J.); sangkyu@knu.ac.kr (S.L.); tlee@knu.ac.kr (T.L.); 2New Drug Development Center, Daegu-Gyeongbuk Medical Innovation Foundation, Daegu 41061, Korea; dha66@dgmif.re.kr (M.J.); jina@dgmif.re.kr (J.K.); 3Mass Spectrometry Based Convergence Research Institute, Kyungpook National University, Daegu 41566, Korea; sunghwank@knu.ac.kr; 4Department of Chemistry, Kyungpook National University, Daegu 41566, Korea; 5Convergence Research Center for Diagnosis, Treatment and Care System of Dementia, Korea Institute of Science and Technology, Seoul 02792, Korea

**Keywords:** estrogen-related receptor γ, inverse agonist, liquid chromatography–tandem mass spectrometry, metabolite identification, metabolomics

## Abstract

DN203368 ((*E*)-3-[1-(4-[4-isopropylpiperazine-1-yl]phenyl) 3-methyl-2-phenylbut-1-en-1-yl] phenol) is a 4-hydroxy tamoxifen analog that is a dual inverse agonist of estrogen-related receptor β/γ (ERRβ/γ). ERRγ is an orphan nuclear receptor that plays an important role in development and homeostasis and holds potential as a novel therapeutic target in metabolic diseases such as diabetes mellitus, obesity, and cancer. ERRβ is also one of the orphan nuclear receptors critical for many biological processes, such as development. We investigated the in vitro metabolism of DN203368 by conventional and metabolomic approaches using high-resolution mass spectrometry. The compound (100 μM) was incubated with rat and human liver microsomes in the presence of NADPH. In the metabolomic approach, the *m/z* value and retention time information obtained from the sample and heat-inactivated control group were statistically evaluated using principal component analysis and orthogonal partial least-squares discriminant analysis. Significant features responsible for group separation were then identified using tandem mass spectra. Seven metabolites of DN203368 were identified in rat liver microsomes and the metabolic pathways include hydroxylation (M1-3), *N*-oxidation (M4), *N*-deisopropylation (M5), *N,N*-dealkylation (M6), and oxidation and dehydrogenation (M7). Only five metabolites (M2, M3, and M5-M7) were detected in human liver microsomes. In the conventional approach using extracted ion monitoring for values of mass increase or decrease by known metabolic reactions, only five metabolites (M1-M5) were found in rat liver microsomes, whereas three metabolites (M2, M3, and M5) were found in human liver microsomes. This study revealed that nontargeted metabolomics combined with high-resolution mass spectrometry and multivariate analysis could be a more efficient tool for drug metabolite identification than the conventional approach. These results might also be useful for understanding the pharmacokinetics and metabolism of DN203368 in animals and humans.

## 1. Introduction

Estrogen-related receptors (ERRs) are orphan nuclear receptors consisting of subunits ERRα, ERRβ, and ERRγ. ERRs are mainly expressed in the brain, heart, kidney, and liver, and they play a vital role in regulating cellular metabolism and energy homeostasis [1]. ERRγ is specifically involved in metabolic diseases such as type 2 diabetes mellitus and alcohol-induced oxidative liver injury [2] caused by impaired hepatic gluconeogenesis [3] and insulin signaling [4]. ERRγ has also been reported to mediate the transcriptional response in cancer [5]. Therefore, it is currently considered a potential therapeutic target for treating metabolic diseases such as type 2 diabetes mellitus and cancer [3,6]. ERRβ has a generic function in regulating energy metabolism like other subtypes; however, it plays a specific role in embryonic development, cell replication and differentiation [7,8]. In particular, ERRβ expression in the breast cancer patient tissues was down-regulated in comparison with normal breast tissues [9]. It suggests that ERRβ may also be a potential therapeutic target.

Several ERRγ-selective inverse agonists have been developed, including GSK5182 [10], DN200434 [11], and DY40 [12]. GSK5182 and DN200434 enhance sodium iodide symporter function and radioiodine activity in anaplastic thyroid cancer cells in vitro, but only DN200434 (which has better in vivo pharmacokinetic profiles and biocompatibility than GSK5182) was effective in an in vivo anaplastic thyroid cancer model. DN203368 ((*E*)-3-[1-(4-[4-isopropylpiperazine-1-yl]phenyl)3-methyl-2-phenylbut-1-en-1-yl]phenol, Figure 1) is a 4-hydroxy tamoxifen analog that is an inverse agonist of estrogen-related receptor β/γ (ERRβ/γ). Development of compound DY40 as an inverse agonist of ERRβ/γ (IC_50_, 0.01 μM) and DY181 as an inverse agonist of ERRβ (IC_50_, 0.05 μM) was reported with only preliminary in vitro data [12]. With the aim of discovering a novel dual ERRβ/γ inverse agonist, a lead compound, DN203368, was developed. Further studies on examining anticancer effects on several in vitro and in vivo cancer models using DN203368 as ERRβ/γ ligand are ongoing.

Drug metabolism is the biotransformation process that makes xenobiotics more polar to facilitate their elimination from the body. The products of metabolism are generally inactive, but active metabolites can also be generated. In some cases, metabolism leads to the formation of reactive or toxic metabolites that can affect drug safety [13]. Drugs such as troglitazone, trovafloxacin, bromfenac, and lumiracoxib have been withdrawn from the market due to adverse effects caused by toxic metabolites [14]. It is thus very important to understand the metabolism of new drug candidates in the early stages of drug discovery and development, and it is also essential for regulatory agencies to verify the safety of the drug.

In general, metabolite identification is conducted by comparing the parent drug’s mass fragmentation pattern with potential metabolites after liquid chromatography–tandem mass spectrometry (LC-MS/MS) [15,16,17]. This conventional approach mainly relies on individual knowledge and experience of drug metabolism and interpretation of mass fragmentation patterns in mass spectra. Metabolomics—the analysis of metabolite populations in various biofluids and tissues—plays an important role in discovering potential biomarker candidates for disease diagnosis. Recently, metabolomics has become a powerful tool for understanding drug metabolism and has been used to identify the metabolic pathways of nintedanib [18], noscapine [19], and PT2385 [20].

This study aimed to evaluate the untargeted metabolomics approach for identifying metabolites of DN203368, a structural analog of 4-hydroxytamoxifen that acts as a dual inverse agonist for ERR β/γ [21], using liquid chromatography with high-resolution mass spectrometry. The evaluation was performed by investigating the metabolism of DN203368 in rat and human liver microsomes, and the findings were compared to the findings of a conventional approach to metabolite identification. Based on the results, we propose a metabolic pathway of DN203368 and demonstrate metabolic differences between species.

## 2. Materials and Methods

### 2.1. Chemicals and Reagents

DN203368, DN203368 *N*-oxide, and *N*-desisopropyl-DN203368 were synthesized by the Daegu-Gyeongbuk Medical Innovation Foundation (Daegu, Korea). Glucose-6-phosphate (G6P), glucose-6-phosphate dehydrogenase (G6PDH), β-nicotinamide adenine dinucleotide phosphate (β-NADP^+^), and magnesium chloride (MgCl_2_) were purchased from Sigma-Aldrich (St. Louis, MO, USA). Pooled human liver microsomes (HLM, catalog No. H2610) and rat liver microsomes (RLM, catalog No. R1000) were purchased from Xenotech (Kansas City, KS, USA). Solvents were high-performance liquid chromatography–mass spectrometry (LC-MS)-grade (Fisher Scientific Co., Pittsburgh, PA, USA), and the other chemicals were of the highest grade available.

### 2.2. Synthesis of DN203368 N-Oxide and N-Desisopropyl-DN203368

(*E*)-4-(4-(1-(3-hydroxyphenyl)-3-methyl-2-phenylbut-1-en-1-yl)phenyl)-1-isopropylpiperazine 1-oxide (DN203368 *N*-oxide). To a solution of (*E*)-3-(1-(4-(4-isopropylpiperazin-1-yl)phenyl)-3-methyl-2-phenylbut-1-en-1-yl)phenol (DN203368) (14 mg, 0.03 mmol) in dichloromethane was added m-CPBA (5 mg, 0.03 mmol) at room temperature. After 10 min, the reaction mixture was quenched with sat. NaHSO_4_ and washed with ethyl acetate. The aqueous layer was neutralized by using sat. NaHCO_3_, extracted with ethyl acetate. The organic layer was dried over Na_2_SO_4_, filtered, and concentrated under reduced pressure. The resulting crude product was purified by column chromatography to obtain DN203368 *N*-oxide (2 mg, 16% yield). MS (ESI^+^) *m**/**z* calculated for C_30_H_37_N_2_O_2_ [M + H]^+^ 456.3; found 456.3. ^1^H NMR (400 MHz, MeOD) δ 7.55 (d, *J* = 8.9 Hz, 2H), 7.13–7.04 (m, 5H), 7.02–6.98 (m, 3H), 6.68 (d, *J* = 7.6 Hz, 1H), 6.63–6.59 (m, 12), 4.39 (t, *J* = 11.6 Hz, 2H), 4.06 (t, *J* = 11.6 Hz, 2H), 3.59–3.51 (m, 1H), 3.13 (t, *J* = 13.8 Hz, 4H), 3.01–2.92 (m, 1H), 1.32 (d, *J* = 6.5 Hz, 6H), 0.85 (d, *J* = 6.9 Hz, 6H). ^13^C NMR (100 MHz, MeOD) δ 157.31 (C), 149.93 (C), 147.30 (C), 144.87 (C), 143.26 (C), 138.66 (C), 137.59 (C), 130.82 (CH), 130.46 (CH), 129.17 (CH), 127.00 (CH), 126.00 (CH), 119.95 (CH), 118.67 (CH), 115.64 (CH), 113.50 (CH), 70.58 (CH), 62.90 (CH_2_), 61.72 (CH_2_), 55.72 (CH_2_), 31.68 (CH), 20.51 (CH_3_), 14.95 (CH_3_).

(*E*)-3-(3-methyl-2-phenyl-1-(4-(piperazin-1-yl)phenyl)but-1-en-1-yl)phenol (*N*-Desisopropyl-DN203368). To a solution of (*E*)-3-(3-methyl-2-phenyl-1-(4-(piperazin-1-yl)phenyl)but-1-en-1-yl)phenyl pivalate (25 mg, 0.05 mmol) in methanol was added potassium carbonate (11 mg, 0.07 mmol). The reaction mixture was stirred at room temperature for 2 h, quenched with distilled water, and the aqueous layer was extracted with ethyl acetate. The combined organic layer was dried over Mg_2_SO_4_, and the solvent was evaporated under reduced pressure. The product was isolated by preparative HPLC to obtain *N*-desisopropyl DN203368 (2.7 mg, 12% yield). MS (ESI^+^) *m/z* calculated for C_27_H_31_N_2_O [M + H]^+^ 399.2; found 399.2. ^1^H NMR (400 MHz, CD_3_OD): δ 7.18 (t, *J* = 8.6 Hz, 3H), 7.11 (td, *J* = 1.2, 8.1 Hz, 3H), 6.80 (dd, *J* = 1.9, 6.8 Hz, 2H), 6.75 (d, *J* = 7.5 Hz, 1H), 6.71–6.69 (m, 2H), 6.58 (d, *J* = 8.8 Hz, 2H), 2.98–2.96 (m, 4H), 2.90–2.88 (m, 4H), 0.95 (d, *J* = 6.9 Hz, 6H).

### 2.3. In Vitro Incubation of DN203368 in Liver Microsomes 

Liver microsomal incubation samples were prepared as described previously [17]. DN203368 (100 μM) was incubated with 1 mg/mL rat or human liver microsomal protein and 100 mM potassium phosphate buffer (pH 7.4) at 37 °C for 5 min. After preincubation, the reaction was initiated by adding an NADPH-generating system (3.3 mM G6P, 1 unit/mL G6PDH, 1.3 mM β-NADP^+^, and 3.3 mM MgCl_2_). The reaction mixtures (final volume 100 μL) were further incubated for 120 min at 37 °C in a heated shaker (Eppendorf, Hamburg, Germany). Samples were prepared in triplicate, and controls comprised heat-denatured microsomal preparations (100 °C for 30 min). The reaction was terminated by adding 100 μL cold acetonitrile followed by centrifugation at 14,000 rpm for 10 min at 4 °C. Finally, the supernatants were concentrated and the residue was reconstituted in 100 μL acetonitrile.

### 2.4. Liquid Chromatography–Tandem Mass Spectrometry (LC-MS/MS)

A Thermo Scientific Vanquish ultra-high-performance liquid chromatography system coupled to a Q Exactive focus orbitrap mass spectrometer (Thermo Fisher Scientific Inc., Waltham, MA, USA) was used to identify DN203368 and its putative metabolites. Chromatography was performed on a Phenomenex Kinetex C18 column (100 × 2.1 mm, 2.6 μm, 100 Å). The mobile phase consisted of water with 0.1% formic acid (A) and acetonitrile with 0.1% formic acid (B). Gradient elution was conducted as follows: 0–1 min, 30% B; 1→5 min, 30%→50% B; 5–7 min, 50% B; 7–7.1 min, 50%→30% B; followed by 3 min re-equilibration (total run time: 10 min). The column oven temperature was maintained at 40 °C. The flow rate was 0.2 mL/min and the injection volume was 2 μL. The electrospray ionization (ESI) parameters were optimized as follows: heated capillary temperature: 320 °C; spray voltage: 3.5 kV; sheath gas flow rate: 40 arb; auxiliary gas flow rate: 10 arb; S-lens RF level: 50.0 V. Nitrogen was used for spray stabilization and as the collision gas in the C-trap. All data were acquired and analyzed using the Thermo Xcalibur 4.0 software (Thermo Fisher Scientific Inc., Waltham, MA, USA).

### 2.5. Metabolite Identification Using the Conventional Approach

For conventional metabolite identification, data were acquired in full scan and parallel reaction monitoring (PRM) mode with an inclusion list of predicted metabolites using liquid chromatography–high-resolution mass spectrometry. The parameters for the full scan mode were as follows: resolution: 70,000; scan range: 300–650; AGC target: 1 × 10^6^; maximum injection time: 100 ms. As for PRM mode: resolution: 17,500; normalized collision energy: 30 eV; AGC target: 5 × 10^4^; maximum injection time 100 ms. An inclusion list contained the precursor ion mass of the predicted metabolic reaction (*m/z* 457.2850 for hydroxylation, *m/z* 473.2799 for dihydroxylation, *m/z* 439.2744 for dehydrogenation, and *m/z* 399.2431 for *N*-deisopropylation), and analysis was performed in positive ion mode. Metabolites were identified by comparing the mass fragmentation pattern of the parent drug with that of potential metabolites.

### 2.6. Metabolite Identification Using a Metabolomic Approach

Data were acquired in full scan and data-dependent MS/MS (ddMS^2^) mode using liquid chromatography–high-resolution mass spectrometry for the metabolomic approach. The full scan mode parameters were set as follows: resolution: 70,000; scan rage: 60–850; AGC target: 1 × 10^6^; maximum injection time: 100 ms. As for ddMS^2^ mode, resolution: 17,500; normalized collision energy: 30 eV, AGC target: 5 × 10^4^; maximum injection time 100 ms. The ddMS^2^ spectra were obtained for the three strongest peaks per cycle in positive ion mode. Mass spectra were processed with Compound Discoverer (version 2.1, 2017, Thermo-Fischer Scientific) to detect peaks, group isotopes, align chromatograms, and generate the ion identity matrix (*m/z* value, retention time, and peak intensity). Results were analyzed by multivariate analysis using SIMCA software (version 14.1, Umetrics, Umeå, Sweden). Principal component analysis (PCA) and orthogonal partial least-squares discriminant analysis (OPLS-DA) were conducted on Pareto-scaled data to identify significant differences between groups. The S-plots generated by OPLS-DA were used to assess relative importance. Potential metabolites were identified by analyzing the ions contributing to the separation of the groups in the S-plots generated by OPLS-DA analysis. DN203368 metabolite structures were identified according to the MS/MS fragmentation patterns compared to the parent drug.

## 3. Results and Discussion

To choose the optimum polarity mode, standard solutions of DN203368 and its metabolites were infused into the mass spectrometer using an electrospray ionization source in positive and negative modes. The results showed that chromatographic peaks in the positive ionization mode were higher than those in negative mode. DN203368 is a 4-hydroxytamoxifen analog with a phenol group, and it is easily protonated in positive ion mode. Therefore, the positive ionization mode was selected, consistent with previous studies [17,22].

### 3.1. DN203368 Metabolite Profiling Using the Conventional Approach

Phase I metabolism of DN203368 was investigated in rat and human liver microsomes using a conventional approach. Following the incubation of DN203368 with RLM and HLM in the presence of an NADPH-generating system, four hydroxylation metabolites (*m/z* 457.2850, M1–M4) and one *N*-deisopropylation metabolite (*m/z* 399.2431, M5) were profiled in PRM mode; dihydroxylation (*m/z* 473.2799) and dehydrogenation (*m/z* 439.2744) metabolites were not observed (Figure 2 and Appendix A). LC-MS/MS analysis of the unchanged DN203368 and its five metabolites produced the informative and prominent product ions for structural characterization (Figure 3 and Figure 4). The structure of DN203368 and its five metabolites were postulated by accurate mass verification and tandem mass spectrum using high-resolution mass spectrometry. The protonated molecular ion of DN203368 was observed at *m/z* 441.2886 with a mass error of −3.17 ppm, eluted at 7.28 min. Typically, the fragment ion at *m/z* 399.2414 was formed via deisopropylation, and the *m/z* 237.1266 ion was formed via loss of a phenyl *N*-isopropylpiperazine group (Figure 3A and Figure 4A). The fragment ion at *m/z* 86.0966 was characteristic of the *N*-ethylisopropyl moiety. The fragment ions of DN203368 were also observed at *m/z* 356.1997 (loss of ethylamine moiety from *m/z* 399.2414), 159.0799 (loss of benzene group and hydrogen molecule from *m/z* 237.1266), 143.0851 (loss of phenol group and hydrogen molecule from *m/z* 237.1266), and 107.0491 (loss of isopropenyl moiety from *m/z* 159.0799) (Figure 3A and Figure 4A).

The protonated molecular ions of M1, M2, M3 and M4 were observed at *m/z* 457.2850 (theoretical *m/z*) and eluted at 5.69, 6.21, 6.42, and 7.46 min, respectively (Figure 2). The accurate mass measurement indicated that the chemical formula was C_30_H_36_N_2_O_2_, suggesting the addition of one oxygen atom to DN203368. These four metabolites (M1–M4) gave characteristic fragment ions necessary for structural identification (Figure 3). The MS/MS spectrum of DN203368 showed a fragment ion at *m/z* 356.1997, whereas M1, M2, and M3 showed a characteristic fragment ion at *m/z* 372.1942. The fragment ions of M1 at *m/z* 159.0801 and 143.0852 were equal to those of the parent, suggesting that the hydroxyl group was introduced to the phenyl moiety in the phenyl *N*-isopropylpiperazine group of DN203368 (Figure 3B and Figure 4B). Fragment ions of M2 and M3 were observed at *m/z* 253.1216, 175.0751, and 123.0441, suggesting hydroxylation of the phenol moiety in the methylphenylbutenyl phenol group of DN203368 (Figure 3C,D and Figure 4C,D). However, the exact hydroxylation site in M2 and M3 could not be determined. 

M4 had a longer retention time than the parent, a characteristic of *N*-oxide [17,23,24]. It was identified as DN203368 *N*-oxide, by co-chromatography and MS/MS spectral data of the authentic synthetic compound (Figure 3E and Figure 4E). The product ion mass spectrum of M4 was different from that of the parent compound; however, its MS/MS spectrum was the same as that of synthesized *N*-oxide standard (Appendix A). The fragment ion at *m/z* 440.2822 was formed via loss of OH, and the *m/z* 425.2587 and 412.2509 were formed via loss of the methylene and ethyl groups from the piperazine ring of *m/z* 440.2822.

The protonated molecular ion of M5 was observed at *m/z* 399.2417, and it was eluted at 6.73 min. M5 was identified as *N*-desisopropyl-DN203368 by co-chromatography and MS/MS spectral data of the authentic synthetic compound (Figure 3F and Figure 4F). M5 exhibited the same fragment ions of DN203368 at *m/z* 356.2005, 267.1479, 237.1266, 159.0799, 143.0851 and 107.0491; however, the fragment ion (*m/z* 86.0966) having an *N*-ethylisopropyl group was not observed due to the loss of isopropyl group. The MS/MS spectrum of M5 indicated that the isopropyl moiety loss occurred in the piperazine ring.

### 3.2. Profiling of DN203368 Metabolites Using a Metabolomic Approach

Compared with the conventional approach, the LC-MS-based metabolomic approach is more effective for the comprehensive profiling of drug metabolites. The multivariate statistical analysis can facilitate drug metabolite characterization and metabolic pathway enrichment [18,25,26]. In this study, a metabolomics approach was applied to characterize drug metabolites of DN203368 comprehensively. PCA analysis showed a clear difference between the DN203368 treatment group and the heat-deactivated control group, as shown in Figure 5A (RLM) and 5D (HLM). *R*^2^X (>0.91) and *Q*^2^ (>0.66) indicate the goodness of fit and predictability of the model, respectively. OPLS-DA analysis also yielded clear separation between the two groups. (RLM (*R*^2^X = 0.955, *Q*^2^ = 1, and *p* = 0.03102) and HLM (*R*^2^X = 0.942, *Q*^2^ = 1, and *p* = 0.02843)). The S-plot of OPLS-DA was used to identify variables contributing to this group separation. Metabolite profiling focused principally on the top-ranking variables, which were observed at the top right position in the S-plot [25,27]. The S-plot revealed several possible DN203368 metabolites and their related ions contributing the most to the separation (Figure 5C,F), which included metabolites found in the conventional method, such as hydroxylated (M1–M3) and *N*-deisopropylated (M5) DN203368 and *N*-oxide metabolite (M4). Two novel metabolites (M6 and M7) produced by unexpected biotransformations were also identified. 

The protonated molecular ion of M6 was observed at *m/z* 415.2729 and eluted at 7.18 min. The accurate mass measurement indicated that the chemical formula was C_28_H_34_N_2_O, suggesting the loss of C_2_H_2_ from DN203368. The fragment ions of M6 at *m/z* 356.2000, 237.1270, 159.0801, 143.0853, and 107.0493 were equal to those of the parent, suggesting that loss of the ethyl group occurred at the isopropylpiperazine group of DN203368 (Figure 3G and Figure 4G). The fragment ion at *m/z* 86.0969 was also observed in the product ion scan mass spectrum of both M6 and DN203368, indicating that the ring opening occurred on the piperazine ring. Based on these results, metabolite M6 was identified as *N*,*N*-desethyl-DN203368. This *N*,*N*-dealkylated metabolite has been reported previously in the metabolism of drugs containing a piperazine ring [12,28].

The protonated molecular ion of M7 was observed at *m/z* 455.2683 and eluted at 7.01 min. The fragment ions of M7 were found at *m/z* 323.1744, 281.1276, and 251.1046, 14 Da greater than those of the parent compound (Figure 3H and Figure 4H). Fragment ions at *m/z* 143.0853 and 86.0968 were also observed in the tandem mass spectrum of both M7 and DN203368. These data indicate that the phenolic ring of DN203368 was hydroxylated followed by oxidation, forming a quinone metabolite. Based on these results, metabolite M7 was identified as the DN203368-quinone. This quinone metabolite was also reported in the metabolism of tamoxifene [29] and toremifene [30], inverse agonists of ERRγ similar to DN203368. According to previous studies, *o*-quinone is a potentially toxic metabolite produced in several compounds, including tamoxifen, toremifene and thalidomide, and *o*-quinone electrophile can react with DNA or protein [31,32,33,34].

### 3.3. Metabolic Pathway and Interspecies Comparison

Based on the identified metabolites, the reaction types, retention times, measured and theoretical masses, mass errors, and chemical formula of DN203368 and its metabolites are summarized in Table 1. The mass errors were within 10 ppm. Seven metabolites (M1–M7) were found in RLM, and five metabolites (M2, M3, M5, M6, and M7) were observed in HLM. *N*-Desisopropyl-DN203369 (M5) was the metabolite with the highest signal intensity in extracted ion chromatograms, and *N*-oxidation was observed, characteristic in RLM. In contrast, HLM primarily utilized hydroxylation (M2 and M3). These results indicate that metabolic differences between species affect the pharmacokinetics of DN203368. The putative metabolic pathway of DN203368 is shown in Figure 6. The observed metabolic pathways were hydroxylation, *N*-oxidation, *N*-dealkylation, and dehydrogenation.

## 4. Conclusions

In this study, seven metabolites (M1–M7) were identified using a metabolomics approach, whereas five metabolites (M1–M5) were identified using an individual knowledge-based conventional approach, indicating that the metabolomic approach is a strong and reliable tool to identify unknown drug metabolites. Seven metabolites of DN203368 in RLM and HLM were structurally characterized using liquid chromatography–high-resolution mass spectrometry. Metabolic pathways included hydroxylation, *N*-oxidation, *N*-dealkylation and dehydrogenation. Comparing the metabolites between species showed that DN203368 *N*-oxide (M4) was rat-specific, and *N*-desisopropyl-DN203368 (M5) was the metabolite with the highest signal intensity in RLM. In contrast, two hydroxylated DN203368 metabolites (M2 and M3) were the metabolites with the highest signal intensity in HLM. The quinone metabolite, regarded as a reactive and toxic intermediate, was found in both liver microsomes. These results might help evaluate the safety of DN203368 and predict the biotransformation and pharmacokinetics in vivo.

## Figures and Tables

**Figure 1 pharmaceutics-13-00776-f001:**
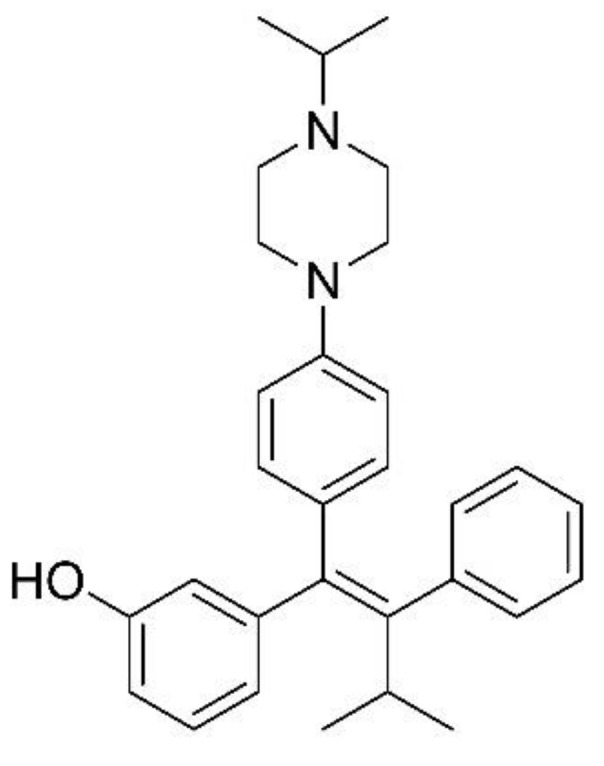
Chemical structure of DN203368.

**Figure 2 pharmaceutics-13-00776-f002:**
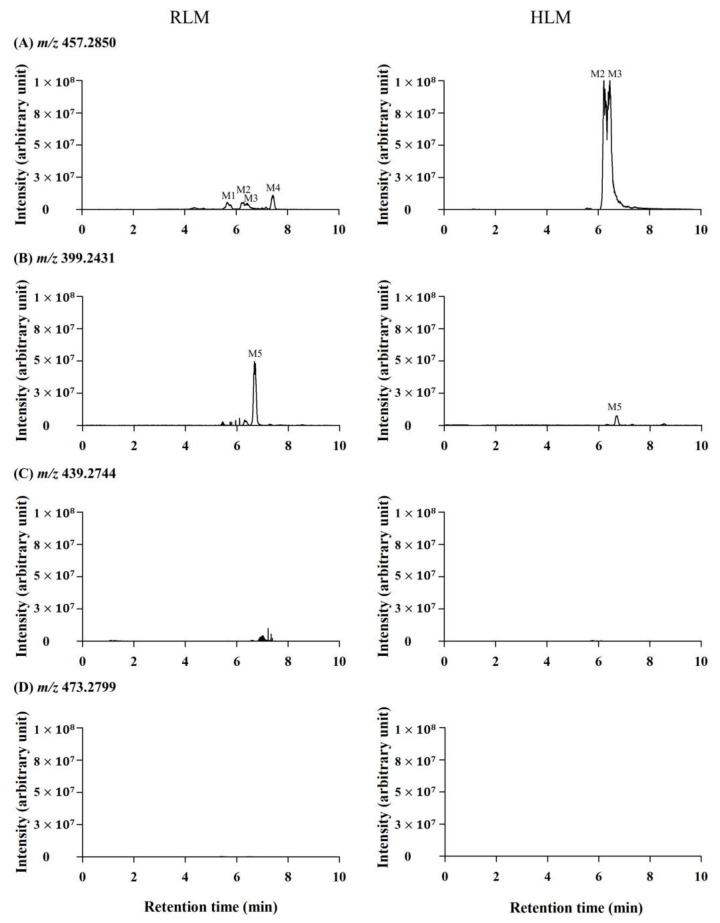
Extracted ion chromatograms (EIC) of DN203368 metabolites obtained from rat and human liver microsomal (RLM, HLM) incubates with DN203368 in the presence of an NADPH-generating system. (**A**) EIC of *m/z* 457.2850 corresponding to the hydroxylation; (**B**) EIC of *m/z* 399.2431 corresponding to the *N*-dealkylation; (**C**) EIC of *m/z* 439.2744 corresponding to the dehydrogenation; (**D**) EIC of *m/z* 473.2799 corresponding to the dihydroxylation.

**Figure 3 pharmaceutics-13-00776-f003:**
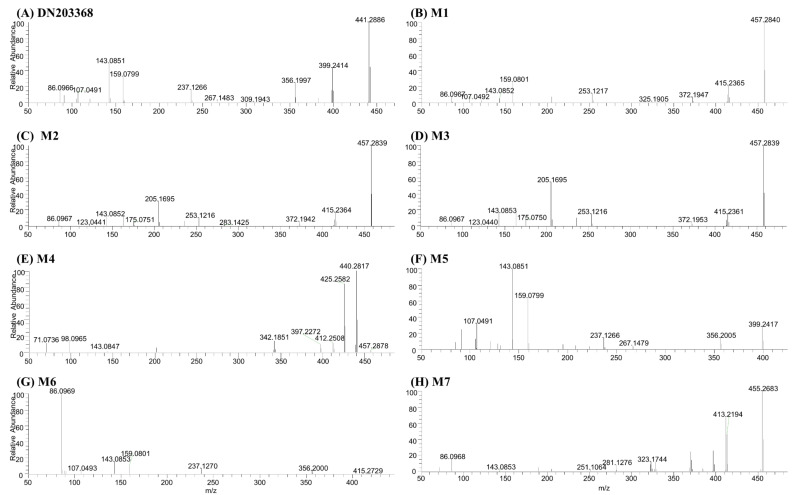
Product ion scan mass spectra of DN203368 (**A**) and its seven metabolites; hydroxy-DN203368 ((**B–D**), M1–M3), DN203368-*N*-oxide ((**E**), M4), *N*-desisopropyl-DN203368 ((**F**), M5), *N*,*N*-desalkyl-DN203368 ((**G**), M6), and DN203368 quinone ((**H**), M7).

**Figure 4 pharmaceutics-13-00776-f004:**
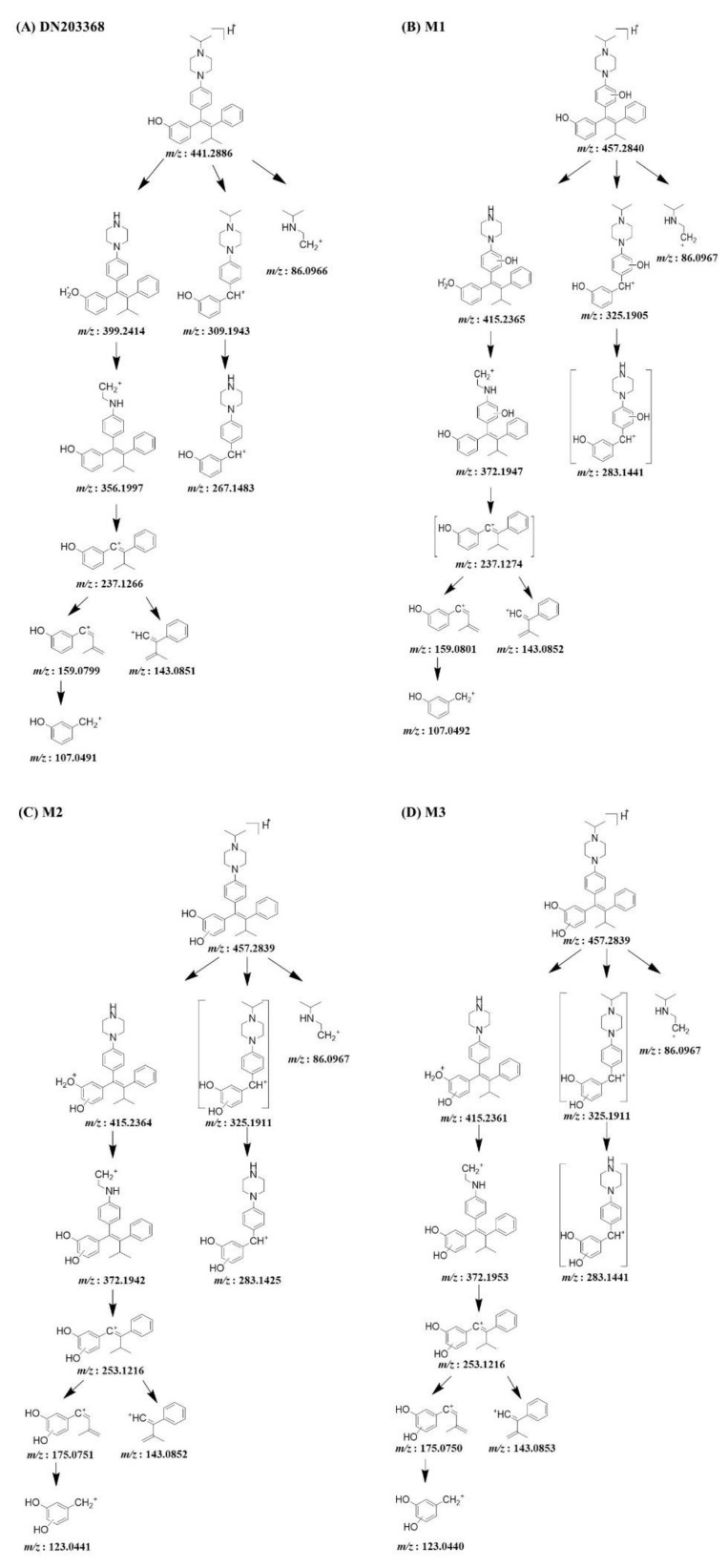
MS/MS fragmentation schemes for DN203368 (**A**) and its seven metabolites; hydroxy-DN203368 ((**B–D**), M1–M3), DN203368-*N*-oxide ((**E**), M4), *N*-desisopropyl-DN203368 ((**F**), M5), *N*,*N*-desalkyl-DN203368 ((**G**), M6), and DN203368 quinone ((**H**), M7).

**Figure 5 pharmaceutics-13-00776-f005:**
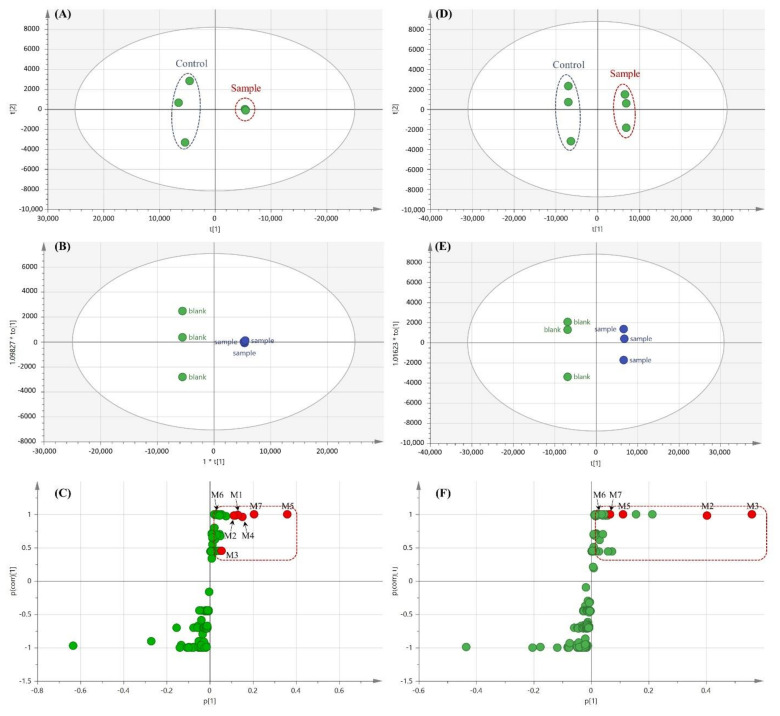
Multivariate data analysis of DN203368 metabolites in rat liver microsomes (RLM, (**A**–**C**)) and human liver microsomes (HLM, (**D**–**F**)). Score plots generated by a principal component analysis (**A**,**D**) and orthogonal partial least-squares discriminant analysis (**B**,**E**); loading S-plot generated by an OPLS-DA for RLM (**C**) and HLM (**F**). The *p*(corr)(1) values represent the interclass difference, and the p(1) values represent the relevant abundance of ions. Data processing and model construction are described in the Materials and Methods. Metabolites of DN203368 are labeled in the S-plot.

**Figure 6 pharmaceutics-13-00776-f006:**
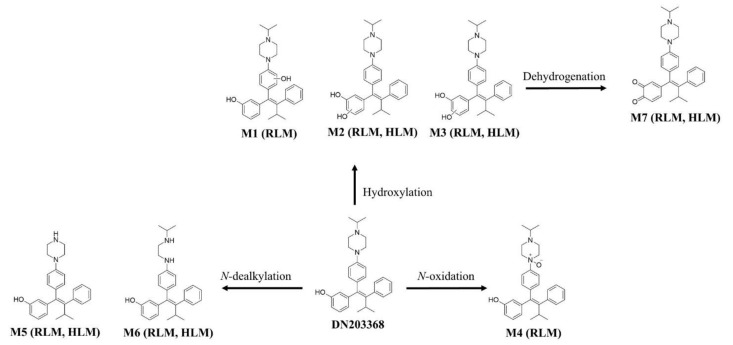
Proposed metabolic pathway of DN203368 in human liver microsomes (HLM) and rat liver microsomes (RLM).

**Table 1 pharmaceutics-13-00776-t001:** Summary of the mass spectral data of DN203368 and its metabolites detected in vitro.

No.	Assignment	*t*_R_ (min)	[M+H]^+^		Error (ppm)	Formula (Neutral)	Conventional Method		Metabolomic Approach	
			Measured	Theoretical			RLMs	HLMs	RLMs	HLMs
M0	DN203368	7.28	441.2886	441.2900	−3.17	C_30_H_36_N_2_O				
M1	Hydroxylation	5.69	457.2840	457.2850	−2.19	C_30_H_36_N_2_O_2_	++		++	
M2	Hydroxylation	6.21	457.2839	457.2850	−2.41	C_30_H_36_N_2_O_2_	++	+++	++	+++
M3	Hydroxylation	6.42	457.2839	457.2850	−2.41	C_30_H_36_N_2_O_2_	++	+++	++	+++
M4	*N*-Oxidation	7.46	457.2863	457.2850	2.84	C_30_H_36_N_2_O_2_	++		++	
M5	*N*-Deisopropylation	6.73	399.2417	399.2431	−3.51	C_27_H_30_N_2_O	++	++	++	++
M6	*N*-Dealkylation	7.18	415.2729	415.2744	−3.61	C_28_H_34_N_2_O			+	+
M7	Hydroxylation, Dehydrogenation	7.01	455.2683	455.2693	−2.20	C_30_H_34_N_2_O_2_			++	++

+++ detected > 10^8^; ++ detected > 10^6^; + detected > 10^4^.

## Data Availability

All data in this study have been included in this manuscript.

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
