# Peer review of "Nontargeted Metabolomics by High-Resolution Mass Spectrometry to Study the In Vitro Metabolism of a Dual Inverse Agonist of Estrogen-Related Receptors β and γ, DN203368"

_pharmaceutics, 2021, doi:10.3390/pharmaceutics13060776_

Round 1

Reviewer 1 Report

In this study, the authors performed both conventional and metabolomic approach to identify the metabolites of Drug DN203368. Overall, the experiment design is reasonable and manuscript writing quality is good. However, the authors should address some issues listed below before considering for publication.

  • Figure quality. The quality of Figure 3 and Figure 4 is not good enough. Please update these two figures. If it is difficult to have several figures in one page, the authors may consider move some figures to supplemental parts.
  • Line 136, could the authors explain why they re-dissolve samples in pure ACN? It is usually good to re-dissolve samples in initial mobile phase. (e.g., 30% B in this study) The peak shape is affected by high content of organic solvents.  
  • It is good to see the metabolomic approach is able to identify more drug metabolites. However, the authors should provide more details here. Theoretically, the control group should not have any drug metabolites been detected. Therefore, the features of M1-M7 should contain 50% of the missing values, at least. What is the missing value filter in this study? How do the authors fill the missing value?

Author Response

Response to Reviewer 1 Comments

Point 1. Figure quality. The quality of Figure 3 and Figure 4 is not good enough. Please update these two figures. If it is difficult to have several figures in one page, the authors may consider move some figures to supplemental parts.

Response 1: Thank you for your kind comment. As you commented, we updated the quality of Figure 3 and Figure 4 by rearranging the components and enlarging figure 4 in our revised manuscript.

Point 2. Line 136, could the authors explain why they re-dissolve samples in pure ACN? It is usually good to re-dissolve samples in initial mobile phase. (e.g., 30% B in this study) The peak shape is affected by high content of organic solvents.

Response 2: Thank you for your valuable comment. In this study, we used 100% acetonitrile to increase the solubility of the nonpolar DN203368 and its metabolites. However, we set the injection volume to 2 mL to minimize the effects of high content of organic solvents.

Point 3. It is good to see the metabolomic approach is able to identify more drug metabolites. However, the authors should provide more details here. Theoretically, the control group should not have any drug metabolites been detected. Therefore, the features of M1-M7 should contain 50% of the missing values, at least. What is the missing value filter in this study? How do the authors fill the missing value?

Response 3: We appreciate your kind comment. Zero imputation was used to replace of corresponding metabolites (variables).

Ref.

Gromski PS et al., Influence of missing values substitutes on multivariate analysis of metabolomics data. Metabolites 2014, 4: 433-452.

Reviewer 2 Report

Nontargeted metabolomics by high-resolution mass spectrometry to study the in vitro metabolism of a new dual inverse agonist of estrogen-related receptor β/γ, DN203368

The current paper describes metabolites, formed in vitro in rat and human liver microsomes from the compound DN203368, which can presumably act as inverse agonist for two estrogen-related receptors beta and gamma. The investigation is made by high-resolution LC-MS with targeted and untargeted approaches, which are correspondingly called ‘conventional’ and ‘metabolomic’ approaches.

Major comments

In the Title, the compound DN203368 is called a ‘new [previously unknown] dual inverse agonist’ compound, while in the Introduction section it is already stated to be [well-known] ‘inverse agonist of estrogen-related receptor’ without any references or proofs. In the Abstract the phrase from the title is softened as ‘which is a potent dual inverse agonist’. Please provide references or proofs of DN203368 action, or soften the title and phases throughout manuscript. The corresponding paragraph in the Introduction should also be rewritten.

The paper describes the advantage of untargeted approach for the discovery of unknown compounds over targeted. For my opinion, untargeted approach with the correct data analysis is a really powerful tool, what is confirmed by this paper. However, the advantage of untargeted approach is not very correctly described here. For the targeted experiment, four masses are set to be detected for the MS instrument in the PRM mode, namely 457.2850, 473.2799, 439.2744, and 399.2431. In the text and in the Table1, it is stated that masses 415.2744 and 455.2693 are not detected. That is quite obvious, because they were not initially set for the detection. If they were set beforehand, they will be detected. The corresponding description of why untargeted approach better should be rewritten.

Authors have revealed and marked out seven metabolites M1-M7 by using untargeted approach and data analysis by oPLS-DA’s S-plot, Figure 5. On this plot, M5 and M7 are the most pronounced for rat model, and M3 and M2 for human model. M1-M7 are marked red. Nevertheless in the S-plot in the upper-right corner there are several other species marked green that are responsible for the separation between two groups – treated with DN203368 and untreated. That is especially pronounced for HLM groups. While M3 has only vague effect in RLM model. Please provide the list of most pronounced compounds (or metabolomic features) marked out by oPLS-DA method, or describe why other compounds are not important.

Authors make conclusions about several species with phrases like ‘most plentiful metabolite’, ‘most abundant metabolite’ and ‘primary products’. These conclusions could only be made based on quantitative data. But firstly, no quantitative data are provided. Plus signs from the Table 1 are not enough; there should be areas of corresponding peaks on EICs; ideally – concentration of metabolites. Secondly, the intensity of the ESI-MS signal is not directly proportional to the compound concentration: compounds with the same concentration in ESI-MS give signal that varies in several orders of magnitude, depending on the nature of the chemicals. Moreover the matrix effects or ionization suppression effects also influence the intensity of compound under study. As summary, one cannot judge about the abundance in complex mixtures based only LC-MS signal intensity/area. Please rewrite such phrases or determine concentrations for all compounds under study.

Figure 2 and line 186. All intensity scales are the same and are provided in absolute units (counts, max. 1e+8). For my opinion, it is better to have percentage scale (like on Fig3), where 100% corresponds to the maximal intensity on the given EIC. As it was already said, in electrospray, the intensity could dramatically change with minor change of the chemical compound. Moreover concentrations of DN203368 derivatives could be too different from the initial DN203368. All that would lead to high intensity differences. Maybe there are something on EICs on Fig 2C and 2D, but we just don’t see in due to inadequate scale?

Paragraph in lines 59-69 describes three compounds that was previously developed by authors. Please state this. Are there any other known or developed ERR inverse agonists found by other groups worldwide? (For example: Yu. DD, et al., doi: 10.1016/j.bmc.2017.01.019; Ghanbari F, doi: 10.1016/j.jsbmb.2019.04.001) It would be beneficial to describe them here in the Introduction as well. Also, there is a description of two more compounds DY40 and DY181, presumably developed by authors as well, with Ref.12. - J Chromatogr B Analyt Technol Biomed Life Sci 2018, 1072, 86-93. But this Ref.12. has nothing common with DY40 and DY181 compounds.

Figure 4 is impossible to read, everything is too small.

Throughout manuscript masses are provided either with four numbers after the decimal point (high-resolution accurate MS, eg. 457.2840) or with one number (low resolution MS, eg. 457.3). It is a bit confusing. Did you measure masses in two modes (high/low res) or two different instruments? Please provide window for the EICs reconstruction. Are these EICs are for low resolution (typically ±0.5 Da), or for high-resolution data? What value ± Da did you use to reconstruct EICs?

Page 9 first paragraph. Coincidence of fragmentation patterns for compounds usually means the coincidence of intensities and masses on MS/MS spectra. M1-M4 have mostly different fragments comparing to DN203368. The difference and coincidence of patterns is well described in the second paragraph for M4, and it is really seen on Fig. S1 and Fig. 3, what I cannot say about M1-M3. The phrase in first paragraph should be rewritten.

Figure 5 presents mixed data from unsupervised PCA and supervised oPLS-DA analyses. Score plot is provided only for PCA, while S-plot is from oPLS-DA. Reader cannot see the clear separation with oPLS-DA method, as the plot is not provided. The score plot for oPLS-DA should be provided additionally to PCA or instead of PCA. Supervised oPLS-DA method on your data should most probably yield clear separation for groups like unsupervised PCA.

Minor comments:

  • After page 6 there are no numbering of lines.
  • Compound DN203368 binds to both ERRβ and ERRγ receptors, thereby in the title it is stated as ‘dual‘ inverse agonist. Then the singular form ‘receptor’ is written. The sentence becomes not very correct. The title should be rewritten to comply language rules, for example ‘…new inverse agonist of both estrogen-related receptors β and γ’ or ‘…new dual inverse agonist of estrogen-related receptors β and γ’ or something similar.
  • Line 16, author affiliation 3 has no address.
  • Lines 61-62, but only DN200434 which has good in vivo → but only DN200434 which has better in vivo
  • Lines 63-64, 4-hydroxy tamoxifen which → 4-hydroxy tamoxifen analog which
  • Line 67, DN2003368 → DN203368
  • Line 137, a remark, sample for HPLC should better be reconstituted in starting conditions of LC program (30% ACN) that should improve resolution.
  • Line 181, Ref 22 has nothing common with the positive ionization MS mode selection
  • Line 189-190. Authors didn’t conduct analysis of elements (C,H,N,O, etc) for molecules. The sentence should be rewritten. Most probably this sentence should describe isotopic pattern or accurate mass verification.
  • Please remove red wavy underlining on Figure S1.
  • Page 9 second paragraph. Repeated sentence should be removed. ‘The MS/MS spectrum of DN203368 showed a fragment ion at m/z 356.2, whereas M1, M2, M3, and M4 showed a characteristic fragment ion at m/z 372.2.’
